# In Vitro Induction of Trained Innate Immunity by bIgG and Whey Protein Extracts

**DOI:** 10.3390/ijms21239077

**Published:** 2020-11-28

**Authors:** Anneke H. Hellinga, Theodoros Tsallis, Talitha Eshuis, Vassilis Triantis, Laurien H. Ulfman, R. J. Joost van Neerven

**Affiliations:** 1FrieslandCampina, 3818 LE Amersfoort, The Netherlands; hellinga.anneke@gmail.com (A.H.H.); pantheotsal@gmail.com (T.T.); talitha.eshuis@frieslandcampina.com (T.E.); vassilis.triantis@frieslandcampina.com (V.T.); laurien.ulfman@frieslandcampina.com (L.H.U.); 2Cell Biology and Immunology, Wageningen University, 6700 AH Wageningen, The Netherlands

**Keywords:** trained immunity, innate immune training, bovine IgG, raw bovine milk, serum protein concentrate, whey protein concentrate, minimal processing, human primary monocytes, infant formula, bioactive component

## Abstract

Bovine immunoglobulin G (bIgG) was previously shown to enhance innate immune responses to toll-like receptor (TLR) stimulation, via induction of trained immunity. In this study, we investigated whether minimally processed dairy streams with high levels of whey proteins as potential infant nutrition ingredients could also induce trained immunity, and to what extent this can be explained by the presence of bIgG. The minimally processed whey ingredients serum protein concentrate (SPC) and whey protein concentrate (WPC) were tested for their ability to induce trained immunity in human peripheral blood monocytes. Both ingredients induced trained immunity as evidenced by an increased production of TNF-α and, to a lesser extent, of IL-6 upon stimulation with TLR ligands. This was comparable to isolated bovine immunoglobulin G (bIgG) that served as positive control. Depletion of bIgG from both whey protein-containing ingredients did not significantly inhibit the induction of trained immunity, suggesting that the streams contain other components in addition to bIgG that are able to induce trained immunity. These results indicate that minimally processed whey ingredients may contribute to protection against infections through enhancing innate immune responsiveness to pathogens.

## 1. Introduction

An infant’s immune system is not yet fully developed in the first year of life. As a result, infants are more vulnerable to infectious diseases, such as respiratory and gastrointestinal tract infections [1]. After birth, breastfeeding protects the infant against infections. A systematic review on 21 studies, performed in industrialized countries, shows that exclusive breastfeeding lowers the risk of overall infections, as well as gastrointestinal and lower respiratory infections [2]. Breastmilk contains a variety of bioactive components that support immune function and provide resistance against infections, but also prevent excessive immune activation that could lead to local inflammation [1,3]. However, when breastfeeding is not possible, milk-based infant formula is used as an alternative for infant feeding [4].

Breastfeeding is the golden standard and ideally, infant formula should provide the same immune support compared to breastfeeding [5]. The most commonly used basis for the production of infant formula is bovine milk [6]. Bovine milk contains many similar immune-related components to breast milk, although concentrations may differ [7]. These bovine milk components have been shown to be functionally active on human immune cells, suggesting they may potentially also provide immune support to infants, when administered in their native form. The bovine whey protein immunoglobulin G (bIgG) has been shown to bind and, in some cases, neutralize a variety of bacterial and viral pathogens. In addition, it can bind to humane myeloid cells via their Fc receptors, leading to phagocytosis and clearance of pathogens [3,8,9]. Interestingly, bIgG has been demonstrated to be able to induce trained immunity, as evidenced by enhanced toll-like receptor (TLR) responsiveness in myeloid cells [10].

“Trained immunity” or “innate immune training” was identified as a mechanism that confers immune memory to the mammalian innate immune system [11]. Until recently, immune memory was thought to reside exclusively in the adaptive immune system. However, Netea and colleagues were the first to describe that innate immune responses towards a primary infection or in vitro stimulation with a pathogen can lead to an enhanced innate immune response to non-related pathogens via innate immune training [11]. This process is independent of the adaptive immune system. Preincubation of peripheral blood mononuclear cell (PBMC)-derived monocytes with bacille Calmette–Guérin (BCG) significantly increased the response of the cells upon restimulation with a range of toll-like receptor (TLR) ligands or heat-killed bacteria as shown by increased production of tumor necrosis factor-α (TNF-α) and interleukin (IL)-6 [12]. Similar results were found for the training of PBMCs and PBMC-derived monocytes with the fungus *C. albicans* and *C. albicans*-derived β-glucan [13]. Training of monocytes led to enhanced production of pro-inflammatory cytokines TNF-α and IL-6, but not anti-inflammatory cytokine IL-10 [13]. As a result, most research on innate immune training focuses on measuring TNF-α and IL-6 as key cytokines affected. This increased innate immune response was observed after re-stimulation with the same stimulus, as well as for re-stimulation by unrelated pathogens, as evidenced by stimulating different TLRs. 

These findings may in part explain why several epidemiological studies have noted non-specific cross-protection against a variety of infectious agents. Vaccination with BCG was repetitively found to be associated not only with protection against tuberculosis, but with an overall decreased incidence of infectious diseases and mortality [14,15,16,17]. More recently, results from a study in elderly people have shown that BCG vaccination protects against respiratory tract infections of probable viral origin [18]. In addition, it has been postulated that induction of trained innate immunity may also protect against emerging viral respiratory infections such as COVID-19 [19], and this concept is currently being investigated in several clinical studies.

Not only vaccines and pathogens (and their derived components) can induce trained immunity, also dietary components have been shown to have the ability to induce trained immunity in vitro [10,20,21,22,23].

Recently, van Splunter et al. studied whether this training effect can also be induced by raw bovine milk or its components [10], as it was already known from epidemiological studies that consumption of raw bovine milk in early life is associated with a lower prevalence of asthma and, in a single study, upper respiratory tract infections [24,25,26]. Training of PBMC-derived monocytes with whey proteins, especially bIgG, resulted in a significantly increased production of pro-inflammatory cytokines IL-6 and TNF-α after re-stimulation with different TLR ligands [10]. This suggests that the functionally intact bIgG may be able to induce a training effect in vivo when present in dairy-based products and may thereby support immune protection against infectious diseases.

Intact bIgG is present in high levels in bovine colostrum (30–200 mg bIgG/mL) and, to a lesser extent, in raw bovine milk (200–500 µg bIgG/mL) [3,27]. Bovine milk is a very commonly used basis for the production of infant formula [6]. However, the production is strictly regulated by the Codex Alimentarius, and therefore raw bovine milk must be processed before it can be included in infant formula [4]. Processing, such as heat treatment, affects the bioactivity and immune activity of proteins [28]. bIgG is thermolabile and after high-temperature processing, the bioactivity and immune activity of bIgG are eradicated in such products. However, bIgG is the most thermostable among the bovine immunoglobulins, and there are still significant levels of intact bIgG present in pasteurized dairy products and mildly processed whey-based food ingredients [29]. 

The aim of the current report was to study if the minimally processed whey protein concentrate (WPC) and serum protein concentrate (SPC) are able to induce trained immunity in human monocytes in vitro, and to study the relative contribution of bIgG in this process.

## 2. Results

### 2.1. Bovine Immunoglobulin G Induces Trained Immunity in Primary Human Monocytes

In order to reproduce the previous finding that bIgG can induce trained immunity, monocytes were isolated from buffy coats, allowed to adhere to the culture plates, and incubated with bIgG, and bovine serum albumin (BSA) as a protein control. After a five-day resting period, the cells were stimulated with the TLR ligands LPS (TLR-4 ligand), R848 (TLR-7 ligand), and Pam3CSK4 (TLR2/TLR1 ligand).

Figure 1A,E show that, when the cells were not stimulated with TLR ligands (control), TNF- α and IL-6 production levels were below the detection level. This was the case for both the trained and non-trained monocytes. Preincubation with bIgG led to increased production of TNF-α and, to a lesser extent, of IL-6 after restimulation with all three TLR ligands (Figure 1B–D,F–H). BSA, as a protein control, did not enhance TNF-α or IL-6 compared to the control. β-glucan that was also used as a positive control, as previously reported by others [13,30,31], only slightly enhanced the production of IL-6 but not of TNF-α (not shown), confirming our own earlier findings [10].

This indicates that the trained immunity induced by bIgG is superior to β-glucan, and is not a non-specific protein effect. For this reason, bIgG, and not β-glucan, was used as control in the next experiments.

As shown in Figure 2A,D, unstimulated monocytes did not produce detectable levels of TNF-α or IL6. After restimulation of the cells that were not trained with bIgG (RPMI) with LPS, R848, and Pam3CSK4, all stimuli induced TNF-α and IL-6 production by the cells (Figure 2B–D,F–H). Training with bIgG, however, resulted in a significantly higher production of TNF-α compared to the untrained cells after re-stimulation with LPS (TLR4 ligand) (Figure 2B), R848 (TLR7/8 ligand) (Figure 2C), and Pam3CSK4 (TLR2/TLR1 ligand) (Figure 2D). In addition, a significantly increased IL-6 production by bIgG-trained monocytes compared to the RPMI control condition was also seen after re-stimulation with LPS (TLR4 ligand) (Figure 2F), but not after re-stimulation with R848 (TLR7/8 ligand) or Pam3CSK4 (TLR2/TLR1 ligand) (Figure 2G,H).

Taken together, these data clearly show the ability of bIgG to induce trained immunity in human monocytes.

### 2.2. Minimally Processed Wheystreams SPC and WPC Induce Trained Immunity in Primary Human Monocytes

To study if mildly processed whey protein ingredient streams that contain bIgG also induce trained innate immunity, the potency of SPC and WPC to induce trained innate immunity was also studied. The SPC and WPC included in the study are minimally processed liquid dairy streams containing mostly whey proteins that have relatively high bIgG concentrations (4.6 and 10.4 mg/mL, respectively). As it was confirmed that bIgG induced trained immunity in monocytes, SPC and WPC were included as a potential training agent in the training experiment. The SPC and WPC were diluted to the same bIgG concentration used in Figure 1, namely 200 µg/mL. The total protein concentration at these dilutions was 7.3 and 2.4 mg/mL for SPC and WPC, respectively.

Figure 3A,E show that training with SPC resulted in a significantly higher production of TNF- α, but not IL-6, compared to the RPMI control condition after re-stimulation with LPS (TLR4 ligand) (Figure 3B,F), R848 (TLR7/8 ligand (Figure 3C,G), and Pam3CSK4 (TLR2/TLR1 ligand) (Figure 3D,H), indicating that SPC was also able to induce trained immunity in monocytes. The data per single donor compared to bIgG and SPC are shown in Appendix A.

Likewise, WPC also resulted in a significantly higher TNF-α production compared to the RPMI control condition after re-stimulation with LPS (TLR4 ligand) (Figure 4B), R848 (TLR7/8 ligand) (Figure 4C), and Pam3CSK4 (TLR2/TLR1 ligand) (Figure 4D). Furthermore, a significantly increased IL-6 production by WPC-trained monocytes compared to the RPMI control condition was measured after re-stimulation with LPS (TLR4 ligand) (Figure 4F), but not after re-stimulation with R848 (TLR7/8 ligand) or Pam3CSK4 (TLR2/TLR1 ligand) (Figure 4G,H). Taken together, similar to bIgG, both SPC and WPC were able to induce trained immunity in monocytes.

### 2.3. Depletion of bIgG from WPC and SPC

In order to study the relative contribution of bIgG that is present in WPC and SPC in innate immune training, both whey streams were depleted of bIgG using magnetic beads coated with protein G.

The depletion of bIgG from different aliquots of WPC and SPC reduced the bIgG concentration by 97.3–99.8%, resulting in bIgG concentrations of 0.7–4.3 µg/mL for WPC and 0.3–0.6 µg/mL for SPC at the final dilution used in in vitro experiments. The depletion procedure reduced the total protein concentration in the WPC and SPC as well (factor 2.2 and 1.6, respectively).

In order to evaluate whether these low bIgG concentrations could still play a role in innate immune training, a titration with these low bIgG (0.2–5 µg/mL compared to 200 µg/mL) concentrations was performed. Figure 5 shows that whereas bIgG induced training at 200 µg/mL, bIgG did not induce training at low concentrations (0.2–5 µg bIgG/mL) in human primary monocytes. These findings indicate that the bIgG concentrations left in the WPC and SPC after depletion are too low to induce trained innate immunity.

### 2.4. Effect of Depletion of Minimally Processed Dairy Streams from bIgG on the Induction of Trained Immunity

To evaluate the relative contribution to the innate immune training effect by SPC and WPC shown in Figure 3 and Figure 4, the effects of depletion of bIgG from SPC and WPC were measured. In this analysis, only donors in which training with SPC or WPC led to an increased TNF-α production of more than 50% compared to RPMI were included. Figure 6A–C and Appendix A show that training with bIgG-depleted SPC-trained monocytes produced similar levels of TNF-α and IL-6 compared to monocytes trained with non-depleted SPC, after re-stimulation with LPS (TLR4 ligand) (Figure 6A), R848 (TLR7/8 ligand) (Figure 6B), and Pam3CSK4 (TLR2/TLR1 ligand) (Figure 6C). Similarly, Figure 6D–E show that depletion of bIgG from WPC also did not lead to a statistically different decrease. However, monocytes trained with bIgG-depleted WPC showed a non-significant reduction in TNF-α production for all stimuli tested.

Taken together, even though bIgG induces innate immune training, depletion of bIgG from WPC and SPC did not result in complete abrogation of innate immune training, suggesting that additional components in minimally processed whey can also induce this effect.

## 3. Discussion

Recently, it was found that raw bovine milk and bIgG were able to induce trained immunity in monocytes [10]. The effect of bIgG occurs at levels that are present in raw milk. Here, we confirm that bIgG induces trained immunity in human primary monocytes and show that the minimally processed whey ingredients SPC and WPC have the same effects on monocytes. Depletion of bIgG from the streams did not reduce (SPC), or only partially reduced (WPC), the training effect, suggesting that even though bIgG induces innate immune training itself, additional whey components can also contribute to the training effect.

The observation that vaccination with BCG (tuberculosis vaccine) also resulted in protection against—and a better innate immune response against—other pathogens has prompted research into the associated mechanisms [15,17].

Interestingly, not only vaccines and pathogens (and components derived from them) can induce trained immunity, but also dietary components have been shown to have the ability to induce trained immunity in vitro [10,20,21,22,23].

In the current manuscript, we confirm these findings for milk-derived bovine IgG and extend these findings to minimally processed whey protein ingredients.

In this study, bIgG, WPC, and SPC induced trained innate immunity as evidenced by increased TNF-α production upon stimulation with TLR-4 (LPS), TLR2/1 (Pam3CSK4), and TLR7/8 (R848) ligands (Figure 1, Figure 2, Figure 3 and Figure 4). The training effect is not simply a non-specific protein effect because the BSA used as a control protein did not induce any training effect.

Depletion of bIgG from WPC and SPC did not prevent the induction of trained immunity. The level of bIgG in the depleted SPC and WPC was considerably lower than the lowest bIgG concentration that could induce trained immunity (Appendix A). bIgG-depleted WPC and SPC could still induce enhanced responses upon TLR stimulation (Figure 5). For WPC, a decrease was noted, but this was not significant. However, this may suggest that although additional components in the whey protein ingredient can also induce trained immunity, for WPC, the relative role of bIgG may be more important than for SPC. Alternatively, as the relative bIgG concentration as part of the total protein concentration was higher in WPC compared to SPC, the high bIgG concentration may offer an alternative explanation for this difference.

The finding that depletion of bIgG does not significantly affect the effect of WPC and SPC implies that bIgG is one of the components present in the ingredients that can induce the effect, but not the only one. We know that many components in raw milk can have an effect on innate immune function [7,32]. In addition, consumption of raw milk is associated with lower prevalence of asthma (for a meta-analysis, see: [33]) but also of upper respiratory tract infections [34], although it should be stated that this is the only publication documenting this to date. Two recent papers are of special interest in relation to innate immune training, one being the publication by van Splunter et al., in which the oral ingestion of bovine lactoferrin induced enhanced cytokine production in PBMCs of elderly women after TLR7/8 stimulation [35], which may suggest training in vivo. This is, however, not supported by other findings that bovine lactoferrin did not induce innate immune training in monocytes of healthy adults in vitro [10]. Such experiments should be repeated using PBMCs of elderly women that have reduced TLR7/8 responses to confirm if this is an explanation. Another recent publication described a role for iron-saturated β-lactoglobulin in the enhancement of quercitin-dependent AhR activation that resulted in decreased antigen presentation in monocytic cells [36]. This suggests that β-lactoglobulin cannot explain our findings as we see enhanced rather than decreased activity of monocytes. Future work should thus focus on trying to identify which whey component(s) in addition to bIgG can induce innate immune training.

Route and time of introduction, as well as the specificity of the trigger inducing the innate immune response, may determine the outcome of a protective or a detrimental effect like chronic inflammation. In addition to bovine milk components, oat-derived (1,3)/(1,4)-β-glucan [20] and oxidized low-density lipoprotein (oxLDL) [21,22], a compound which is increasingly present in the blood as a result of a Western diet, have also been shown to induce trained immunity.

BCG vaccination is applied intradermally and early in life and induces a systemic response that does more than only protect against tuberculosis. In fact, an epidemiological study showed that BCG vaccination led to an overall decrease in the incidence of infections, morbidity, and mortality [14]. Further in vitro and in vivo experiments with mice indeed revealed that vaccination with BCG led to an increased innate immune response upon heterologous secondary stimulation compared to the non-vaccinated control [12,30,37].

Beta glucans and bovine milk components are dietary components that may have local effects in the intestine. For this, sufficient amounts of intact and bioactive bIgG must be able to survive stomach digestion and become available to the immunological tissues in the small intestine. A recent review on the survival of orally administered bIgG found that recovery of bIgG in the stool of infants ranges from 4 to 13% [38]. In addition, in an in vitro digestion model mimicking digestion in the infant intestine, more than 50% of bIgG and bIgA survived digestion until the end of the small intestine, in contrast to less than 20% in an adult digestion model (unpublished observations). This clearly indicates that indeed part of the ingested bIgG survives infant digestion throughout the entire gastrointestinal tract and thus is available to the immunological tissues in the small intestine. Stomach digestion in infants is less stringent compared to adult stomach digestion, due to its higher pH [39], and more degradation of bIgG in adults has been shown. This would mean that bIgG-induced innate immunity is more likely to play a role in infant than in adult life. Further, it is known that oral intake of bIgG does not induce the uptake of bIgG into the circulation [40].

However, IgG from human breast milk has been shown to be bidirectionally transported over the intestinal epithelium by binding to the neonatal Fc receptor, which is expressed on intestinal epithelial cells throughout life in humans [29,41]. bIgG has also been shown to be able to bind human FcRn, although FcRn has a higher affinity for human IgG [42,43]. Non-digested bIgG can thus bind to CD32 [8] as well as to FcRn on myeloid and epithelial cells. Intestinal dendritic cells can extrude their dendrites into the gut lumen to capture antigens [44] and proteins including bIgG via Fc receptors. In addition, luminal dietary proteins, like bIgG, can be transferred via the M cells, thus reaching myeloid immune cells that reside in the Peyer’s patch [45]. This suggests that bIgG, similarly to human IgG [29,41], can be transported by epithelial cells, via dendrites of antigen-presenting cells, and via M cells from the gut lumen into the mucosa or Peyer’s patches where it can exert its effect on immune cells (macrophages, DC).

As bovine milk components and oat-derived glucans are introduced in early life and may act locally in the intestine rather than systemically, a contribution to the protection against infection is expected. Actually, consumption of raw bovine milk in early life has been associated with lower prevalence of airway infections and asthma later in life [24,25,26,34]. It is tempting to speculate that milk components in early life contribute to this protective effect via innate immune training. Moreover, a recently published in vitro study shows that the training effect of BCG between monocytes from adults versus cord blood monocytes revealed that monocytes of a newborn, trained with BCG, have similar cytokine production after re-stimulation, compared to adult monocytes [37]. This advocates that trained immunity may be a mechanism to compensate for the lack of adaptive immune response in early life. Whether bovine milk components induce a similar monocyte response in cord blood versus adult blood compared to BCG remains to be investigated. As infants, especially the ones who are not breastfed, have a higher risk for common childhood infections, application of the concept of trained immunity in infant formula might provide protection against infectious diseases for formula-fed infants.

Additional studies are needed to directly show an effect on enhanced innate immune function after oral delivery of bIgG or the whey ingredients SPC or WPC. The data shown here suggest that these components, when used in infant formula, may contribute to the protection against infectious diseases by inducing trained innate immunity.

## 4. Materials and Methods

### 4.1. Minimally Processed Dairy Streams

Two minimally processed whey ingredients were provided by FrieslandCampina InnovationCentre (Wageningen, The Netherlands). These were Vivinal MFGM, an MFGM-rich whey protein concentrate, further referred to as WPC, and native whey protein concentrate, further referred to as SPC. WPC has a total protein concentration of 127 mg/mL, of which 10.43 mg/mL is bIgG. SPC has a total protein concentration of 169 mg/mL, of which 0.46 mg/mL is bIgG. Endotoxin levels of the streams were measured by performing Endolisa^®^ according to the manufacturer’s protocol (Hyglos, cat. #609033).

### 4.2. Isolation of Bovine IgG

Bovine IgG (bIgG) was isolated from the raw colostrum. For this, first a whey concentrate was prepared from the raw colostrum. The raw colostrum was diluted 1:3 with simulated milk ultrafiltrate. The lipid fraction was removed by centrifugation at 8000 rpm for 20 min at 4 °C. Then, the caseins were precipitated by adding 1 M HCl to pH 4.2. The precipitated casein was separated and removed from the solution by centrifugation at 12,000 rpm for 20 min at 20 °C. The concentrated whey was adjusted to pH 6.8 with 1 M NaOH, and then filtered with a 0.45 µm filter to remove debris. The bIgG was purified from the colostrum whey concentrate with a protein G column (Sigma Aldrich, St. Louis, MO, USA, cat. #29-0485-81), according to the manufacturer’s protocol.

### 4.3. Depletion of Minimally Processed Dairy Streams from bIgG

SPC and WPC were 2.5× and 5× diluted with Dulbecco’s Phosphate Buffered Saline (DPBS) (Gibco, Waltham, MA, USA, cat. #14190-144), respectively, and filtered (Corning, cat. #431219) to remove aggregates and debris. Bovine IgG depletion was performed with protein G magnetic beads (Thermo Scientific, Waltham, MA, USA, cat. #88848). An amount of 800 µL of beads was washed by gently vortexing with 200 µL DPBS in a 1.5 mL Eppendorf tube. The supernatant was removed by placing the tube in a magnetic stand (Invitrogen, cat. #12321D), enabling pipetting the liquid from the tube. The beads were then washed again with 1 mL DPBS in a similar manner. After washing, 100 µL 2.5× diluted SPC or 5× diluted WPC were incubated with the beads in a thermomixer at 1000 rpm at room temperature (RT). The supernatant (=depleted SPC or WPC) was collected from the tube in the magnetic stand. The protein concentration of the SPC and WPC samples was measured before and after depletion with the BCA Protein Assay Kit (ThermoFisher, Waltham, MA, USA, cat. #23225) according to the manufacturer’s protocol with BSA used as reference protein.

### 4.4. bIgG Quantification in Depleted Dairy Stream and Isolated bIgG Samples

The bIgG concentration in depleted dairy stream and isolated bIgG samples was quantified with the ELISA Quantitation Set according to the manufacturer’s protocol (Bethyl, Montgomery, TX, USA, cat. #E10-118). A 96-well plate (Corning, Corning, NY, USA, cat. #3590) was coated by 1-h incubation with anti-bIgG antibody (Bethyl, cat. #A10-118A) diluted in 0.05 M carbonate-bicarbonate coating buffer (Bethyl, cat. #E107). After 1-h incubation, the plate was washed five times with 0.05 M Tris buffered Saline with Tween (Sigma Aldrich, cat. #T9039) dissolved in demineralized water according to the manufacturer’s protocol. Then, 200 µL/well blocking solution (Roche, Basel, Switzerland, cat. #11112589001) was added and incubated for 30 min at RT. The wells were again washed five times. The depleted dairy stream and isolated bIgG samples, with undiluted and 2.5× diluted SPC and undiluted and 5× diluted WPC as controls, were diluted 100×, 1000×, 10,000×, and 100,000× in blocking buffer. The diluted samples and controls, and standard (Bethyl, cat. #RS10-104-5) were added at 100 µL/well in duplicate and incubated for 1 h at RT. The wells were washed again five times. The horseradish peroxidase (HRP) detection antibody (Bethyl, A10-118P) was added at 100 µL/well and incubated for 1 h at RT. The wells were washed again five times. TMB (at RT) (ThermoFisher, cat. #SB02) was added at 100 µL/well and incubated for 10–15 min. The color reaction was stopped by adding 1 M HCl (Merck, Kenilworth, NJ, USA, cat. #1.09057.1000) at 100 µL/well. The absorbance was measured on an iMark absorbance microplate reader (Bio-Rad, Hercules, CA, USA, cat. #12757) set to 450 nm with 655 nm as correction.

### 4.5. Training Experiment with Human Primary Monocytes

The training experiment was performed according to the protocol described in (Figure 7) [10]. However, a 2-h adherence step was introduced as it was observed that the number of adhered monocytes was significantly lower for the wells with the dairy stream samples, compared to the RPMI control condition.

#### 4.5.1. Training Agents

The training agents SPC, WPC, SPC- and WPC-isolated bIgG, and colostrum-isolated bIgG (positive control) were included in the training experiment in a concentration of 200 µg bIgG/mL, similar to the concentration of bIgG present in raw bovine milk [3]. The depleted SPC and WPC were diluted equivalent to non-depleted SPC and WPC, respectively. B-glucan was extracted from *C. albicans* and diluted to 1 µg/mL. BSA (Roche, cat. #10735094001) was added as a protein control and diluted to 7 mg/mL. The batches of SPC and WPC and depleted WPC and SPC were tested for endotoxin levels. The tested samples contained less than 0.5 ng endotoxins/mL at the concentration used in the training experiment, which is considered not to affect the monocytes [46]. The BSA concentration was based on the protein concentration of the dairy stream with the highest protein concentration, SPC. All training agents were diluted in RPMI, supplemented with 1%penicillin/streptomycin (pen/strep) mix (ThermoFisher, cat. #15140122). RPMI 1%pen/strep was also a negative control for training. The training agents and RPMI control were added in the absence of serum.

#### 4.5.2. Training Experiment

Primary human monocytes were isolated from fresh buffy coats from the Sanquin bloodbank (Nijmegen, The Netherlands). The buffy coats were diluted 1:3 with DPBS (Gibco, cat. #14190-144) supplemented with 2% pooled fetal bovine serum (FBS). PBMCs were isolated with SepMate tubes (Stemcell, Vancouver, BC, Canada, cat. #85450) according to the manufacturer’s protocol, with Ficoll-Paque PLUS (GE Healthcare, Chicago, IL, USA, cat. #17144002) as the density gradient medium. Monocytes were isolated from the PBMCs by negative selection with the EasySep isolation kit according to the manufacturer’s protocol (Stemcell, cat. #19359). Cell count of PBMCs and monocytes was performed with a Scepter cell counter according to the manufacturer’s protocol (Merck, cat. #PHCC20040), with a cut-off set to a cell diameter of 5.708–13.15 µm. As there were differences in monocyte yield per donor, not all test groups could be taken along for all donors tested, resulting in a slight difference in the numbers of donors tested per condition.

Human primary monocytes were taken up in RPMI 1640 medium (ThermoFisher, cat. #A1049101), supplemented with 1% pen/strep, to a concentration of 10^6 monocytes/mL and added to a 100 µL/well 96-well plate (Corning, cat. #3585). The monocytes were incubated for 2 h at 37 °C 5% CO_2_ to let the monocytes adhere to the bottom of the well. After 2 h, 100 µL/well warm 2× concentrated training agents was added to the monocytes. The monocytes were incubated with the training agents for 24 h at 37 °C 5% CO_2_. After 24 h of training, the training agents were removed by washing with warm DPBS and subsequently with warm DPBS 2% FBS. Then, warm RPMI, supplemented with 1% pen/strep mix and 10% human pooled serum, was added to the monocytes. After two or three days, the medium was refreshed with warm RPMI, supplemented with 1% pen/strep mix and 10% human pooled serum (Sigma Aldrich, cat. #H3667). On day 6, the cells were re-stimulated by different TLR ligands: Pam3CSK4 (1 µg/mL) (Invivogen, San Diego, CA, USA, cat. #tlrl-pms), LPS (0.1 µg/mL) (Sigma Aldrich, cat. #L6529), and R848 (10 µg/mL) (Invivogen, tlrl-r848), targeting TLR2/1, TLR4, and TLR7/8, respectively. RPMI, supplemented with 1% pen/strep, was included as a negative control for re-stimulation. The cells were incubated for 24 h at 37 °C 5% CO_2_. After 24 h, the supernatant was collected and stored at −20 °C for the IL-6 and TNF-α concentration measurement.

#### 4.5.3. Cytokine Concentration in Supernatant

The concentration of pro-inflammatory cytokines IL-6 and TNF-α was assessed with Human IL-6 DuoSet ELISA (R&D Systems, Minneapolis, MN, USA, cat. #DY206) and Human TNF-α DuoSet ELISA (R&D Systems, cat. #D210) according to the manufacturer’s protocol. The ELISAs for both cytokines were performed in duplicate. The supernatant was incubated on a capture antibody-coated plate. After incubation with a HRP-conjugated antibody, TMB (ThermoFisher, cat. #SB02) was added to induce color development. The color reaction was stopped by adding 1 M HCl (Merck, cat. #1.09057.1000). The absorbance was measured with an iMark absorbance microplate reader (Bio-Rad, cat. #12757) set to 450 nm with 560 nm as correction.

#### 4.5.4. Statistical Analysis

Statistical analysis was performed using GraphPad Prism version 8.4.2. The training effect was assessed by comparing the cytokine concentrations of each training agent with the negative control of the same donor and the re-stimulation conditions. Normality of residuals was assessed for each comparison with the Anderson–Darling, D’Agostino–Pearson omnibus, Shapiro–Wilk, and Kolmogorov–Smirnov tests. Residuals were considered normally distributed if more than two of the aforementioned tests resulted in *p* < 0.05. Comparisons which were normally distributed were tested for significance with the paired t test (*p* < 0.05). Comparisons which were not normally distributed were tested for significance with the Wilcoxon matched-pairs signed rank test (*p* < 0.05).

## 5. Conclusions

In conclusion, this study confirms the finding that bovine IgG induces trained immunity in primary human monocytes in vitro. In addition, both mildly processed whey ingredients WPC and SPC induced trained immunity as evidenced by an increased production of TNF-α and, to a lesser extent, of IL-6 upon stimulation with TLR ligands. As depletion of bIgG from the whey ingredients did not significantly inhibit the induction of trained immunity, not only bIgG but also additional whey components play a role in the induction of trained immunity. These results indicate that minimally processed whey ingredients may contribute to protection against infections through enhancing innate immune responsiveness to pathogens.

## Figures and Tables

**Figure 1 ijms-21-09077-f001:**
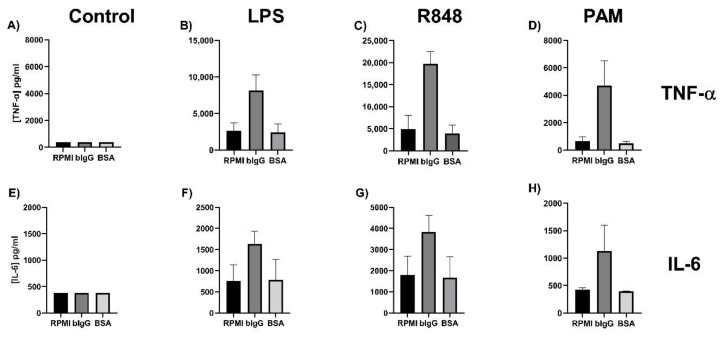
Induction of trained immunity by bovine immunoglobulin G (bIgG). Monocytes were isolated from fresh buffy coats of three independent donors. After allowing the monocytes to adhere for 2 h, the training stimuli bIgG (200 µg/mL), β-glucan (1 µg/mL), and protein control bovine serum albumin (7 mg/mL) and the negative control (only culture medium RPMI) were added. After 24-h incubation with the stimuli, the monocytes were washed to remove the stimuli. On day 6, the monocytes were re-stimulated with (**A**,**E**) RPMI without serum, (**B**,**F**) LPS (100 ng/mL), (**C**,**G**) R848(10 µg/mL), or (**D**,**H**) PAM (Pam3CSK4, 10 µg/mL). After 24-h incubation, the supernatant was collected for assessment of (**A**–**D**) TNF-α and (**E**–**H**) IL-6 production. The lowest level of detection is 390 and 375 pg/mL for TNF-α and IL-6, respectively.

**Figure 2 ijms-21-09077-f002:**
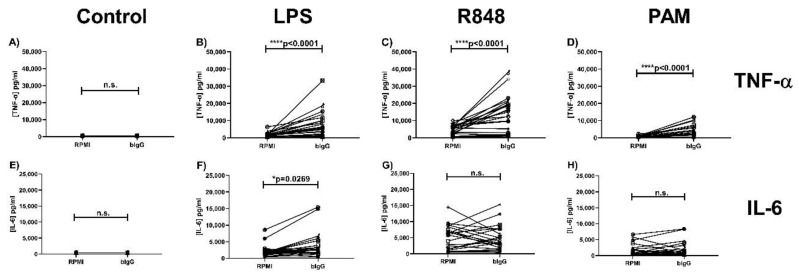
Induction of trained immunity in primary human monocytes by bIgG. Monocytes were isolated from buffy coats and after allowing the monocytes to adhere for 2 h, the training stimulus bIgG (200 µg/mL) and the control RPMI were added. After 24-h incubation with the stimulus, the monocytes were washed to remove the stimuli. On day 6, the monocytes were re-stimulated with (**A**,**E**) RPMI without serum, (**B**,**F**) LPS (100 ng/mL), (**C**,**G**) R848 (10 µg/mL), or (**D**,**H**) PAM (Pam3CSK4, 10 µg/mL). After 24-h incubation, the supernatant was collected for assessment of (**A**–**D**) TNF-α and (**E**–**H**) IL-6 production in duplicate The lowest level of detection is 390 and 375 pg/mL for TNF-α and IL-6, respectively. * *p* < 0.05, **** *p* < 0.0001.

**Figure 3 ijms-21-09077-f003:**
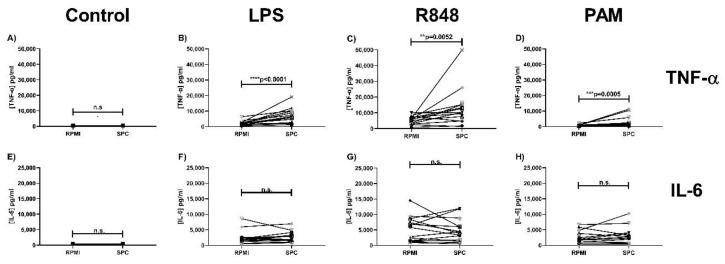
Induction of trained immunity by serum protein concentrate (SPC). Monocytes were isolated from buffy coats and after allowing the monocytes to adhere for 2 h, the training stimulus SPC (200 µg bIgG/mL) and the negative control RPMI were added. After 24-h incubation with the stimulus, the monocytes were washed to remove the stimuli. On day 6, the monocytes were re-stimulated with (**A**,**E**) RPMI without serum, (**B**,**F**) LPS (100 ng/mL), (**C**,**G**) R848 (10 µg/mL), or (**D**,**H**) PAM (Pam3CSK4, 10 µg/mL). After 24-h incubation, the supernatant was collected for assessment of (**A**–**D**) TNF-α and (**E**–**H**) IL-6 production. ** *p* < 0.01, *** *p* < 0.001, **** *p* < 0.0001.

**Figure 4 ijms-21-09077-f004:**
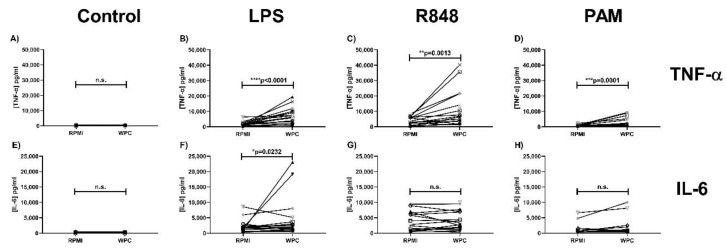
Whey protein concentrate (WPC) induces trained immunity in primary human monocytes. Monocytes were isolated from fresh buffy coats and after allowing the monocytes to adhere for 2 h, the training stimulus WPC (200 µg bIgG/mL) and the control RPMI were added. After 24-h incubation with the stimulus, the monocytes were washed to remove the stimuli. On day 6, the monocytes were re-stimulated with (**A**,**E**) RPMI without serum, (**B**,**F**) LPS (100 ng/mL), (**C**,**G**) R848(10 µg/mL), or (**D**,**H**) PAM (Pam3CSK4, 10 µg/mL). After 24-h incubation, the supernatant was collected for assessment of (**A**–**D**) TNF-α and (**E**–**H**) IL-6 production. N = 17, except for Pam3CSK4 re-stimulation, where N = 14. * *p* < 0.05, ** *p* < 0.01, *** *p* < 0.001, **** *p* < 0.0001.

**Figure 5 ijms-21-09077-f005:**
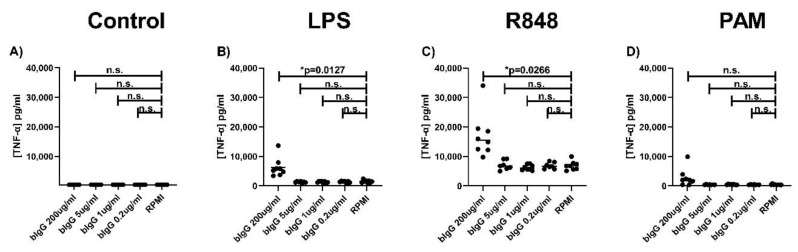
Bovine immunoglobulin does not induce trained immunity in human primary monocytes at lower concentrations. Monocytes were isolated from fresh buffy coats and after allowing the monocytes to adhere for 2 h, the training stimulus bIgG (5, 1 and 0.2 µg/mL) and the control RPMI were added. After 24-h incubation with the stimulus, the monocytes were washed to remove the stimuli. On day 6, the monocytes were re-stimulated with (**A**) RPMI without serum, (**B**) LPS (100 ng/mL), (**C**) R848 (10 µg/mL), or (**D**) PAM (Pam3CSK4, 10 µg/mL). After 24-h incubation, the supernatant was collected for assessment of TNF-α production (**A**–**D**). The lowest level of detection is 390 and 375 pg/mL for TNF-α and IL-6, respectively. N = 7–9. * *p* < 0.05.

**Figure 6 ijms-21-09077-f006:**
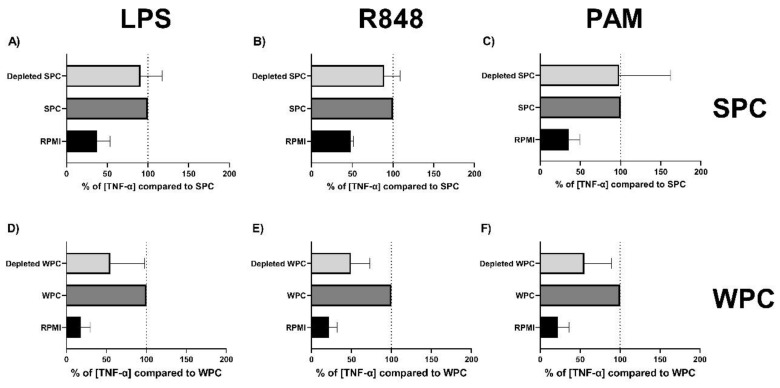
Effect of depletion of bIgG from SPC and WPC on the induction of trained immunity in primary monocytes. Monocytes were isolated from fresh buffy coats and after allowing the monocytes to adhere for 2 h, the training stimuli depleted SPC (diluted to the dilution corresponding to 200 µg bIgG/mL non-depleted SPC) and depleted WPC (diluted eq. to 200 µg bIgG/mL non-depleted WPC) and the negative control RPMI were added. After 24-h incubation with the stimulus, the monocytes were washed to remove the stimuli. On day 6, the monocytes were re-stimulated with (**A**,**D**) LPS (100 ng/mL), (**B**,**E**) R848 (10 µg/mL), or (**C**,**F**) PAM (Pam3CSK4, 10 µg/mL). After 24-h incubation, the supernatant was collected and TNF-α production was measured. Data are shown as % TNF-α production relative to the non-depleted SPC and WPC. Error bars represent the standard deviation.

**Figure 7 ijms-21-09077-f007:**
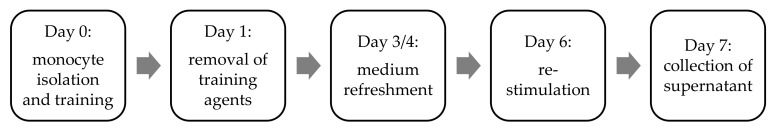
Overview of the training experiment. At day 0, monocytes were isolated from fresh buffy coats and after allowing the monocytes to adhere for 2 h, the training stimuli bIgG (200 µg/mL), β-glucan (1 µg/mL), protein control BSA (7 mg/mL), SPC (200 µg bIgG/mL), depleted SPC (diluted eq. to 200 µg bIgG/mL non-depleted SPC), WPC (200 µg bIgG/mL), and depleted WPC (diluted eq. to 200 µg bIgG/mL non-depleted WPC) and the negative control RPMI were added. After 24-h incubation with the stimulus, the monocytes were washed to remove the stimuli. The monocytes were allowed to rest for five days with a medium refreshment on day 3 or 4. On day 6, the monocytes were re-stimulated with LPS (100 ng/mL), R848 (10 µg/mL), or PAM (Pam3CSK4, 10 µg/mL). After 24-h incubation, the supernatant was collected for the assessment of IL-6 and TNF-α production.

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
