# Peer review of "In Vitro Induction of Trained Innate Immunity by bIgG and Whey Protein Extracts"

_ijms, 2020, doi:10.3390/ijms21239077_

Round 1

Reviewer 1 Report

Review IJMS 989041

Hellinga et al in their review concentrate of trained immunity, a recent topic of interest considering the participation of innate immunity in defence.

The concept to exploit cow’s milk as an important dietary substance for this goal is interesting and the methods used are suitable for a proof of concept.

The authors investigate bovine IgG from serum protein concentrate and whey protein concentrate.  Basically, the results indicate that this is not the case. Overall thus, though it is interesting to explore, the data indicate that other compounds than bovine Igs do the job.

Major Comments:

Several of my comments while reading were answered a few lines later by the authors, e.g. Is the level of bovine IgG in milk actually high enough to expect trained immunity? Does this mirror the stimulation concentration of 200μg bIgG/ml? Yes, indeed.

In other words, would the assumption be practical? Yes: 200-500μg bIgG/ml are contained in raw milk.

There are 2 hypotheses how bovine Ig could work: a) via the variable domains, in this case the bovine IgG needs to be specific for human infectious agents in order to be protective. This is not likely.

  1. b) via FcgammaR´s. In studying the interaction of bovine Igs with human Fc receptors, the authors took as control BSA.

Qu1: why didn´t they use human IgG – would this be the most appropriate control?

Qu2: Why did they look exactly at TNF-a and IL-6? Why didn’t they check for IL-10 as a control?

Overall, the data propose that the streams contain other components in addition to bIgG that are able to induce trained immunity. Problem: The authors themselves interpret the results controversary:

  1. In the current manuscript, we confirm these findings for milk-derived bovine IgG, and extend these findings to minimally processed whey protein ingredients.

OR:  B) Depletion of bIgG from WPC and SPC did not prevent the induction of trained immunity.

  • They should be congruent in the result that Igs do not do the job, but something else, probably whey proteins. This must be properly discussed. There is recent data on the immunomodulatory potency of whey protein beta-lactoglobulin (Roth-Walter, doi: 10.1016/j.jaci.2020.05.023.

In this respect, I also wonder why the authors have not added recent data on a study with minimally processed cows milk showing effects in allergy as well as in infections (Brick et al, DOI: 10.1016/j.jaip.2019.11.017), and the data by Roth-Walter et al which indicate a role for beta-Lactoglobulin as the major whey protein compound. This should be added to the discussion in order to relieve the reader from the frustration that there is no answer.

Minor comments to improve the ms.

  • To my opinion, the discussion on oxLDL has nothing to do that the study and the whole § should be removed as follows: With respect to route, LDL is oxidized in the body in the vascular endothelium, ….
  • Similarly, the comparisons to the innate training effects of BCG have maybe place in the introduction, but this has nothing to do with the present study, and should be omitted from the discussion. I find it more systematic to stick on the dietary aspects.
  • How should the uptake of bovine Ig be? They say, that orally delivered bIgG showed an effect on enhanced innate immune function. But there is a conflict as the authors state that oral intake of bIgG does not induce the uptake of bIgG into the circulation. On the other hand, relation to human milk is an impracticable concept.

Author Response

Review IJMS 989041 Hellinga et al

Dear reviewers, editor,

First of all, we would like to thank the reviewers for reading and commenting on our manuscript. Please find below a point by point reply to the issues raised. Changes in the text of the manuscript itself are marked by track changes, and referred to in our replies.

Please note that numbered references in the replies refer to the reference numbers in the revised manuscript.

Sincerely

Joost van Neerven

Reviewer 1 

Hellinga et al in their review concentrate of trained immunity, a recent topic of interest considering the participation of innate immunity in defense.

The concept to exploit cow’s milk as an important dietary substance for this goal is interesting and the methods used are suitable for a proof of concept.

The authors investigate bovine IgG from serum protein concentrate and whey protein concentrate.  Basically, the results indicate that this is not the case. Overall thus, though it is interesting to explore, the data indicate that other compounds than bovine Igs do the job.

Major Comments:

Several of my comments while reading were answered a few lines later by the authors, e.g. Is the level of bovine IgG in milk actually high enough to expect trained immunity? Does this mirror the stimulation concentration of 200μg bIgG/ml? Yes, indeed.

In other words, would the assumption be practical? Yes: 200-500μg bIgG/ml are contained in raw milk.

Comment:  we have made this point more explicit by adding the sentence “The effect of bIgG occurs at levels that are present in raw milk.” to the discussion, second sentence of the discussion on p8.

In fact in very recent experiments we have also titrated bIgG in the higher concentration ranges and noted that after 100-75 ug/ml the effect of bIgG starts to decrease, meaning that even in pasteurized milk IgG levels are high enough (that is around 100-200 ug/ml depending on heat/temperature profile used). Please see the attachment titration bIgG.pdf for your reference wherein each series is a single donor.

There are 2 hypotheses how bovine Ig could work: a) via the variable domains, in this case the bovine IgG needs to be specific for human infectious agents in order to be protective. This is not likely.

  1. b) via FcgammaR´s. In studying the interaction of bovine Igs with human Fc receptors, the authors took as control BSA.

Qu1: why didn´t they use human IgG – would this be the most appropriate control?

Reply: The suggestion of using human IgG as a control is very interesting, and in another research line we are actually planning to look if human IgG has a similar effect to understand the relevance of these findings, and to understand if indeed CD32 has a role in the process as we expect based on (den Hartog et al 2014, REF #8). However in this case we wanted to use a protein control of a protein present in bovine whey preparations, and for this we quite arbitrarily chose BSA as an easily available purified protein that is present in bovine whey. The fact that bovine BSA is not having an effect shows at least that the effect of bovine IgG is not a random bovine protein induced effect.

Qu2: Why did they look exactly at TNF-a and IL-6? Why didn’t they check for IL-10 as a control?

Reply: In the first trained immunity paper in 2012, Quintin et al (Quintin et al Cell Host Microbe 2012 REF # 13) studied the effect of b-glucan induced trained immunity, and in this study b-glucan was shown to induce trained immunity by induction of TNF-a and IL-6, but not IL-10. As a result, most research on innate immune training focuses on these cytokines as a read out and not on IL-10 (REF #12 on BCG; and Saeed Science 2014 on b-glucan). In collaboration with the Nijmegen group we used these cytokines as read out for innate immune training of bIgG, which worked well  as we reported in Van Splunter et al Nutrients 2018. In this work we also measured IL-10 but also in our hands IL-10 was not informative.

To illustrate this, we have changed and added a sentence in the introduction:

The sentence: ”Training of monocytes led to enhanced production of pro-inflammatory cytokines TNF-α and IL-6, but not anti-inflammatory cytokine IL-10” on page 2  was changed into: “Training of monocytes led to enhanced production of pro-inflammatory cytokines TNF-α and IL-6, but not anti-inflammatory cytokine IL-10  [13]. As a result, most research on innate immune training focuses on measuring TNF-α and IL-6 as key cytokines affected.”

Overall, the data propose that the streams contain other components in addition to bIgG that are able to induce trained immunity. Problem: The authors themselves interpret the results controversary:

  1. In the current manuscript, we confirm these findings for milk-derived bovine IgG, and extend these findings to minimally processed whey protein ingredients.

OR:  B) Depletion of bIgG from WPC and SPC did not prevent the induction of trained immunity.

  • They should be congruent in the result that Igs do not do the job, but something else, probably whey proteins. This must be properly discussed. There is recent data on the immunomodulatory potency of whey protein beta-lactoglobulin (Roth-Walter, doi: 10.1016/j.jaci.2020.05.023.

In this respect, I also wonder why the authors have not added recent data on a study with minimally processed cows milk showing effects in allergy as well as in infections (Brick et al, DOI: 10.1016/j.jaip.2019.11.017), and the data by Roth-Walter et al which indicate a role for beta-Lactoglobulin as the major whey protein compound. This should be added to the discussion in order to relieve the reader from the frustration that there is no answer.

Reply: The reply to this comment is twofold, on the one hand we show that bovine IgG, that is present at sufficient concentrations in the whey ingredients has the ability to induce trained immunity at these concentrations, indicating it does induce innate immune training. We depleted bIgG to check if bIgG can completely explain the effect of the ingredients. We found out that removal of bIgG does not significantly inhibit the effect, although there is some decrease in IgG-depleted WPC as illustrated in figure 6. This clearly suggests that in addition to bIgG there must be another component(s) in whey ingredients that can also induce trained immunity. To add clarity to the manuscript we have addressed   in the text by inserting the following paragraph in the discussion p8-9, including reference to the Roth Walter and Brick and additional papers :

“The finding that depletion of bIgG does not significantly affect the effect of WPC and SPC implies that bIgG is one of the components present in the ingredients that can induce the effect, but not the only one. We know that many components in raw milk can have an effect on innate immune function [7, 32]. In addition, consumption of raw milk is associated with lower prevalence of asthma [for meta analysis see: 33] but also of upper respiratory tract infections [34], although it should be stated that this is the only publication documenting this to date. Two recent papers are of special interest in relation to innate immune training, the publication by van Splunter et al in which the oral ingestion of bovine lactoferrin induced enhanced cytokine production in PBMC of elderly women after TLR7/8 stimulation [35], which may suggest training in vivo. This is, however, not supported by other findings that bovine lactoferrin did not induce innate immune training in monocytes of healthy adults in vitro [10]. Such experiments should be repeated using PBMC of elderly women that have reduced TLR7/8 responses to confirm if this is an explanation. Another recent publication described a role for iron saturated b-lactoglobulin in the enhancement of quercitin-dependent AhR activation that resulted in decreased antigen presentation in monocytic cells [36]. This suggests that b-lactoglobulin cannot explain our findings as we see enhanced rather than decreased activity of monocytes. Future work should thus focus on trying to identify which whey component(s) in addition to bIgG can induce innate immune training.”

Minor comments to improve the ms.

  • To my opinion, the discussion on oxLDL has nothing to do that the study and the whole paragraph should be removed as follows: With respect to route, LDL is oxidized in the body in the vascular endothelium, ….

Reply: We have removed this section as indicated by the reviewer.

  • Similarly, the comparisons to the innate training effects of BCG have maybe place in the introduction, but this has nothing to do with the present study, and should be omitted from the discussion. I find it more systematic to stick on the dietary aspects.

Reply: We have removed most of this section, and moved the sentence “More recently, results from a study in elderly people have shown that BCG vaccination protects against respiratory tract infections of probable viral origin [18]. In addition, it has been postulated that induction of trained innate immunity may also protect against emerging viral respiratory infections such as COVID-19 [19], and this concept is currently being investigated in several clinical studies.“ to the introduction, page 2.

How should the uptake of bovine Ig be? They say, that orally delivered bIgG showed an effect on enhanced innate immune function. But there is a conflict as the authors state that oral intake of bIgG does not induce the uptake of bIgG into the circulation. On the other hand, relation to human milk is an impracticable concept.

Reply: Thank you for the comment. We hypothesize that bIgG works locally in the gut but not systemically. This is based on the following data. As to uptake of bovine IgG, we do know that bIgG is not taken up into the serum of adults. On the other hand we know from work by (Den Hartog et al 2014, ref #8) that bIgG can bind to FcgR on human monocytes and other immune cells. Intestinal dendritic cells can extrude their dendrites into the gut lumen [44] and proteins , like bIgG, can be transferred via the M cells reaching myeloid immune cells in the Peyer’s patch [45]. In this way, the bIgG in the gut lumen can come into contact with myeloid immune cells, probably followed by immediate internalization as described by den Hartog et al, thus not becoming available in the blood, but being in direct contact with myeloid cells of the immune system. This is of course depending on the bIgG being intact (see comment to reviewer # 3)

To illustrate this we changed:

 “bIgG has also been shown to be able to bind human FcRn, although FcRn has a higher affinity for human IgG [38,39]. This suggests that bIgG, similarly to human IgG [27,37], can be transported over the intestinal epithelium, into the mucosa or Peyer’s patches where it can exert its effect on immune cells (macrophages, DC) in the lamina propria.”

 into:

Non-digested bIgG can thus bind to CD32 [8] as well as to FcRn on myeloid and epithelial cells.  Intestinal dendritic cells can extrude their dendrites into the gut lumen to capture antigens [44] and proteins including bIgG via Fc receptors. In addition luminal dietary proteins, like bIgG, can be transferred via the M cells, thus reaching myeloid immune cells that reside in the Peyer’s patch [45]. This suggests that bIgG, similarly to human IgG [29,41], can be transported by epithelial cells, via dendrites of antigen presenting cells, and via M cells from the gut lumen into the mucosa or Peyer’s patches where it can exert its effect on immune cells (macrophages, DC). “ – page 9

Reviewer 2 Report

The authors pose the scientific question if bovine IgG or whey protein do affect the human immune response, respectively the innate immune training.

The idea is to support whey proteins as potential infant nutrition ingredients. It were evidenced by increased production of TNF-alpha and IL-6. The activity was tested with a series of TLR ligands.

As control experiments the authors depleted bIgG, which did not impact the the levels of TNF-alpha and IL-6.

As this work is of interest to a broader audience, the authors should rethink the way the wrote the paper and should make it a bit less technical. I ask you to add a graphical abstract. In any case the authors use acronyms in the abstract, please change this.  

The weekly training experiment: Why a week, is this related in any way to digestion?

Is this a citation: Van Splunter and colleagues (2018)? Then it lacks completeness, else, who is Van Splunter? Please check your script it is filled with typos. 

Nice work

Author Response

Review IJMS 989041 Hellinga et al

Dear reviewers, editor,

First of all, we would like to thank the reviewers for reading and commenting on our manuscript. Please find below a point by point reply to the issues raised. Changes in the text of the manuscript itself are marked by track changes, and referred to in our replies.

Please note that numbered references in the replies refer to the reference numbers in the revised manuscript.

Sincerely

Joost van Neerven

Reviewer 2

The authors pose the scientific question if bovine IgG or whey protein do affect the human immune response, respectively the innate immune training.The idea is to support whey proteins as potential infant nutrition ingredients. It were evidenced by increased production of TNF-alpha and IL-6. The activity was tested with a series of TLR ligands.

As control experiments the authors depleted bIgG, which did not impact the levels of TNF-alpha and IL-6.As this work is of interest to a broader audience, the authors should rethink the way the wrote the paper and should make it a bit less technical. I ask you to add a graphical abstract. In any case the authors use acronyms in the abstract, please change this.  

Reply: We added a graphical abstract (see attached PDF file) and have changed the abstract as requested. We trust that the changes in the manuscript will make the story a bit easier for a broader audience.

The weekly training experiment: Why a week, is this related in any way to digestion?

Reply: No, this is the standard innate immune training assay as regularly used. This is not related to digestion. Actually the training agent is reduced by washing after an incubation period of 2 hours, so the cells return to a resting state before restimulation after 6 days

Is this a citation: Van Splunter and colleagues (2018)? Then it lacks completeness, else, who is Van Splunter? Please check your script it is filled with typos. 

Reply: The reviewer is right, there were many typos that we initially overlooked, we have located them and corrected them, as we did with the referral to the paper by van Splunter et al. that was changed into: “in [10]’on page 11, section 4.5

Nice work

Reply: Thank you very much !

Reviewer 3 Report

Dear Editor,

the manuscript entitled "In vitro induction of trained innate immunity by bIgG and whey protein extracts" by Hellinga et al. reports about the efficacy of bovine IgG in milk derivatives serum protein concentrate (SPC) and whey protein concentrate (WPC) in inducing trained immune response through production of infammatory cytochines. The study is thoroughly presented and the findings are interesting in order to elucidating molecular aspects of healthy benefits of breast milk feeding. The work opens several questions which remain unanswered, like the effect of digestion on bIgG, which kind of products can be originated and which are the milk components responsible for the trained immunity induction afetr bIgG depletion.

I think these unanswered questions could be experimentally faced in a future work, but it would be useful to provide a deeper argumentation about these aspects in the Discussion section of the paper.

In vitro digestion's simulation of bIgG could be informative about the biological impact of this study.

Some little typos are present throughout the manuscript.

Author Response

Dear reviewers, editor,

First of all, we would like to thank the reviewers for reading and commenting on our manuscript. Please find below a point by point reply to the issues raised. Changes in the text of the manuscript itself are marked by track changes, and referred to in our replies.

Please note that numbered references in the replies refer to the reference numbers in the revised manuscript.

Sincerely

Joost van Neerven

Reviewer 3

Dear Editor,

the manuscript entitled "In vitro induction of trained innate immunity by bIgG and whey protein extracts" by Hellinga et al. reports about the efficacy of bovine IgG in milk derivatives serum protein concentrate (SPC) and whey protein concentrate (WPC) in inducing trained immune response through production of infammatory cytokines. The study is thoroughly presented and the findings are interesting in order to elucidating molecular aspects of healthy benefits of breast milk feeding. The work opens several questions which remain unanswered, like the effect of digestion on bIgG, which kind of products can be originated and which are the milk components responsible for the trained immunity induction after bIgG depletion.

Comment: we added the phrase on page 8: “The effect of bIgG occurs at levels that are present in raw milk.” At the beginning of the discussion. This means that any product containing 200 ug/ml IgG can have an effect. As indicated in the reply to reviewer 1, we now know that even at lower concentrations IgG is effective (75-100 ug/ml and even a bit lower). So pasteurized milk can also apply. As to powdered products like milk powder but also infant nutrition, bIgG is denatured and is not present at these high concentrations yet. We have deliberately not included this in the discussion to stay away from products, but to focus on the immunological effects, we hope you can agree with this.

I think these unanswered questions could be experimentally faced in a future work, but it would be useful to provide a deeper argumentation about these aspects in the Discussion section of the paper.

In vitro digestion's simulation of bIgG could be informative about the biological impact of this study.

Reply: We agree with the reviewer that this would be very informative. The text refers to digestion in this section: “A recent review on the survival of orally administered bIgG found that recovery of bIgG in the stool of infants range from 4-13% [34]. This clearly indicates that indeed part of the ingested bIgG survives infant digestion throughout the entire gastrointestinal tract and thus is available to the immunological tissues in the small intestine. The stomach digestion in infants is less stringent compared to adult stomach digestion, due to its higher pH [35] and more degradation of bIgG in adults has been shown This would mean that bIgG induced innate immunity is more likely to play a role in infant than in adult life. Also, it is known that oral intake of bIgG does not induce the uptake of bIgG into the circulation [36]. “

In addition, we have performed digestions in house, in which we have seen that approximately 60% of bovine milk IgG and IgA survive digestion until the end of the small intestine in an infant digestion model, and only approximately 20% if you mimic the adult digestion.  To address this we have inserted the sentence on page 9:  “In addition, in an in vitro digestion model mimicking digestion in the infant intestine more than 50% of bIgG and bIgA survived digestion until the end of the small intestine, in contrast to less than 20% in an adult digestion model (unpublished observations)”.

For your reference please see the attached PDF file.

Some little typos are present throughout the manuscript.

Reply: Thank you, we have corrected them in the revised manuscript.

Round 2

Reviewer 1 Report

I have no further comments.